# Mechanistic prediction of community composition across resource conditions and species richness

Zhijie Zhang[1,2] ✉ & Lutz Becks [2]

Predicting species coexistence and community assembly is a central goal in ecology. Traditional methods, based on the effect of one species on another (e.g., Lotka-Volterra competition model), are sensitive to environmental context. This is because they ignore the fundamental processes that can be applied across environments. While mechanistic approaches offer promise, empirical tests remain rare. Here, we integrate a mechanistic consumer-resource model with the growth of 12 phytoplankton species in monoculture over a range of nitrate, ammonium or phosphorous concentrations. We find that the mechanistic approach accurately predicts the composition of 960 communities across species richness and resource conditions. We confirm by simulations, species competing for substitutable resources (nitrate vs. ammonium) exhibit greater diversity than those competing for essential resources (nitrate vs. phosphorus), especially when initial species richness is high. This is because, in competition for essential resources, each species is likely to consume less of the resource that is more limiting to its growth, which violates the mechanistic rule of coexistence that states that each species must consume more of the resource that more limit it. Our study highlights the power of the mechanistic approach in understanding and predicting species loss across environments and, ultimately, mitigating its pace.

Understanding and predicting species coexistence often relies on the Lotka[1] and Volterra[2] framework. This approach, inferring competitive interactions from the effects of one species on the growth of another, has provided numerous insights. For example, it has revealed the relative strength of intra- to interspecific competition as a major determinant of species coexistence[3] and has been used to predict community composition[4,5]. However, without explicitly identifying the causes of competition, it fails to capture the dependence of species coexistence on environments, such as resource availability[6–8] and species richness[9], limiting its applications. Consequently, mechanistic approaches are much needed. One such approach, formularized by MacArthur, describes how species consume and convert shared resources and thus compete with each other[10,11]. Expanding upon this groundwork, two

rules for species coexistence were identified by Tilman[12,13] (Fig. 1a). First, each species must be limited by different resources. Otherwise, a 'superior' species with the lowest resource requirements will outcompete others. Second, each species must consume more of the resource that more limits itself. Despite the enduring interest in this mechanistic approach, direct empirical studies are rare[14,15]. The few empirical studies focus on resource requirement[16] or the effect of resource availability on species coexistence, leaving the important role of resource consumption largely unexplored (but see refs. 17–19). Furthermore, most studies focus on species pairs and a single type of resource (e.g., only essential[17] or substitutable[18] resources). A comprehensive understanding of the proportion of species that meet Tilman's two rules for coexistence remains elusive.

[1]State Key Laboratory for Vegetation Structure, Function and Construction (VegLab), Institute of Ecology, College of Urban and Environmental Science, Peking University, Beijing, China. [2]Aquatic Ecology and Evolution, University of Konstanz, Konstanz, Germany. ✉e-mail: zhijie.zhang@pku.edu.cn

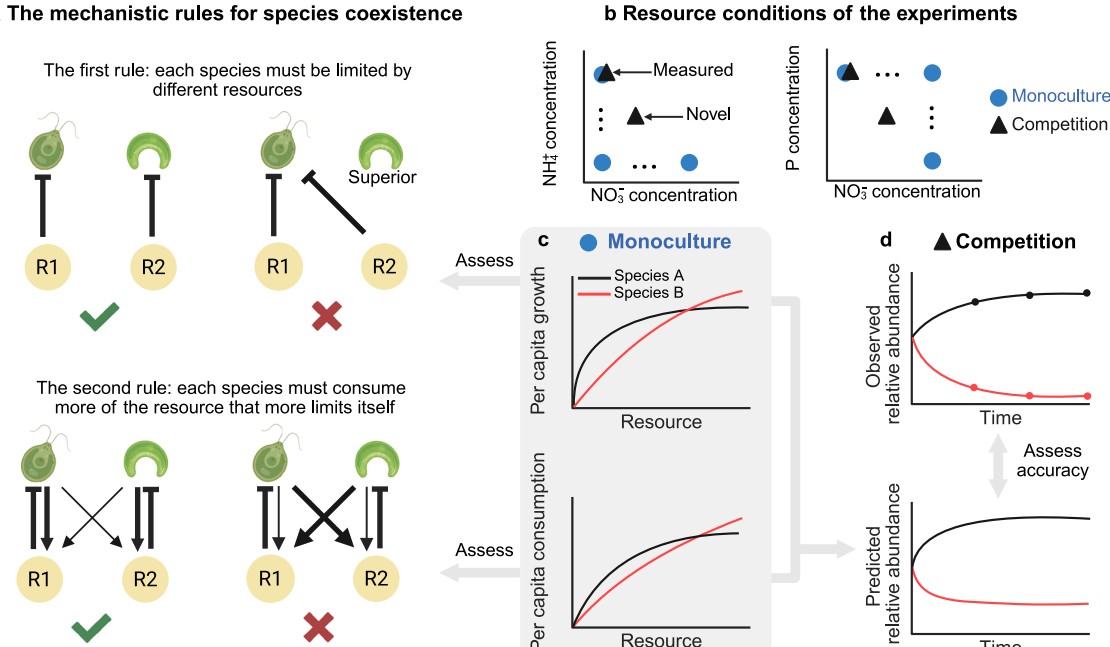

**Fig. 1 | A mechanistic approach to understanding and predicting species coexistence. a** Tilman's rules for species coexistence, as illustrated by two algae consuming two resources (See Supplement Note 4 for the corresponding Zero Net Growth Isoclines [ZNGI] figures). First, each species must be limited by different resources. Second, they must consume more of the resource that more limits itself. Inhibitors indicate the limitation of resources on algal growth (resource requirement). Arrows indicate the effect of algae on resources (resource consumption), with thicker arrows representing a larger effect. **b** In a monoculture experiment, each of the 12 algal species was grown under 12 concentrations of nitrate ($NO_3^-$), ammonium ($NH_4^+$), or phosphorus (P). In a competition experiment, two, three, four, or six species were competed for either two substitutable resources ($NO_3^-$ and $NH_4^+$) or essential resources ($NO_3^-$ and P). Measured conditions refer to resource conditions used in both monoculture and competition experiments, while novel conditions are those used only in the latter. **c** To quantify resource requirement and consumption, we combined a consumer-resource model with the monoculture experiment. We further assessed the proportion of species that met Tilman's mechanistic rules. **d** We predicted community composition and assessed predictive accuracy with the competition experiments. Fig. 1a was created in in BioRender. Becks, L. (2025) https://BioRender.com/oyxxims.

Here, we test 1) whether the mechanistic approach can well-predict community composition and 2) to what extent the two rules for coexistence described by Tilman are fulfilled. We used unicellular phytoplankton (freshwater green algae) as the study system. Their small size and fast generation time provide a controllable, rapid and scalable system to test theoretical models. Unlike bacteria − where cross-feeding, or the sharing of metabolites between species, is common[20] − phytoplankton primarily compete for resources, such as light, nitrogen, and phosphorus and descriptions of interactions between phytoplankton species via metabolites are limited to toxins[21].

We track the daily growth rates of 12 different phytoplankton species (Table S1) in monocultures over four days (approx. zero to eight generations, depending on the resource concentrations). This is conducted under 12 concentrations of nitrate ($NO_3^-$), ammonium ($NH_4^+$), or phosphorus (P; Fig. 1b), while ensuring that all other nutrients remained unlimited. To quantify the resource requirement and consumption for each species, we use Bayesian modeling to parameterize a consumer-resource model using the growth data and initial resource concentrations (see Materials and Methods for details; Fig. 1c). Next, we grow two, three, four, or six species in semi-continuous cultures where they compete for different ratios of two essential resources ($NO_3^-$ and P) or of two substitutable resources ($NO_3^-$ and $NH_4^+$; Figs. 1b, S1, 2). We track the community composition over 12 days with an automated pipeline that integrates high-content microscopy, imaging analysis, and machine learning; and compare the observed composition of these 960 communities with predictions from the consumer-resource model and parameters derived from the monocultures (Fig. 1d). Finally, we assess the proportion of communities meeting Tilman's two rules for coexistence using our experiments and simulation.

## Results and Discussion
### The mechanistic approach predicts community composition
The per capita growth rate and consumption rate increased asymptotically with resource concentration for all 12 algal species (Fig. 2), indicating that all species can grow in monocultures, although with a large variation among species (Fig. 2). Using this information on resource requirement and consumption, we asked whether the mechanistic, resource-consumer approach can predict the composition (as species relative abundance) of 960 communities where species competed for essential or substitutable resources. For comparison, we predicted relative abundance using a null model in which species relative abundances were randomly shuffled within each community to be predicted. This model predicted the community composition with a mean accuracy of 53.5% (Fig. 3a; assessed by the Bray-Curtis similarity between the predicted and observed species relative abundance). Then, we predicted relative abundance with the consumer-resource model. In this model, for essential resources ($NO_3^-$ and P), we assumed that the growth of a species was determined by the resource that supports the lower growth rate, according to the Liebig's law of minimum[22]. For substitutable resources ($NO_3^-$ and $NH_4^+$), we assumed that the growth rate is the sum of the growth rates provided by both resources. Although the relative abundance strongly depended on resource condition and changed over time (Figs. S4, 5; Supplement Note 1), it was well-predicted by the mechanistic model, with a mean accuracy of 83.4% (Figs. 3a, S6; Table S2; Linear mixed-effects model (LMM): $F_{1, 14251} = 2656$, $P < 0.001$).

Our mechanistic approach had also robust predictive abilities across resource conditions. It accurately predicted community composition of the 960 communities, not only in measured conditions, in which resource requirement and consumption were quantified from monocultures, but also in novel conditions (i.e., combinations N ad P

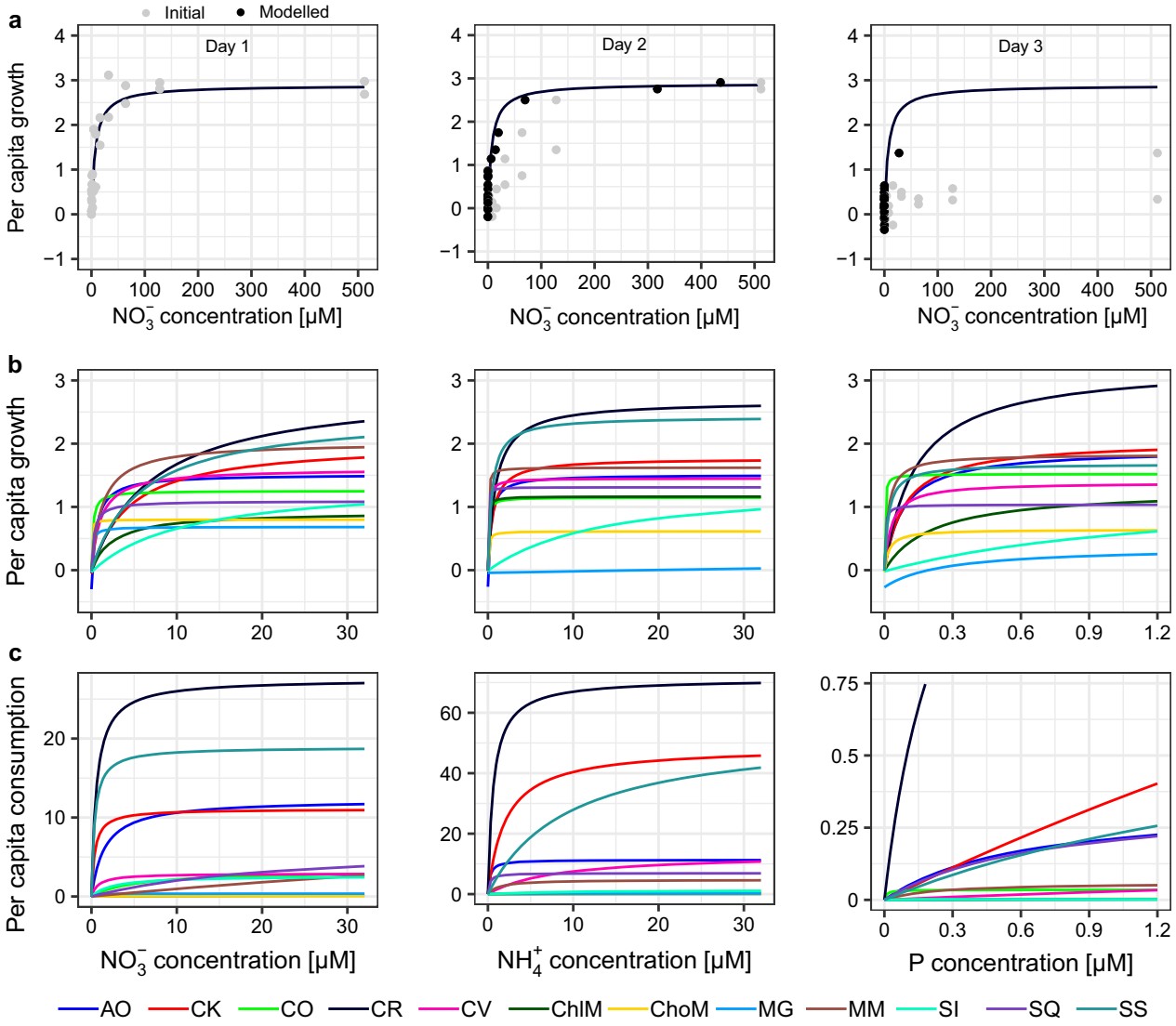

**Fig. 2 | The 12 algal species vary in resource requirement and consumption.**
**a** Using time series data, the consumer-resource model captured both resource requirement and consumption of *Chlamydomonas reinhardtii* on nitrate ($NO_3^-$; see Fig. S3 for all species and all three resources). On day 1, the fitted resource requirement curve explained the relationship between growth rate and initial $NO_3^-$ concentration (gray dots). On days 2 and 3, using the initial concentration always overestimated the growth rate because it ignored resource consumption. The modeled $NO_3^-$ concentration (black dots), which accounted for resource consumption, better explained the growth rate. **b, c** The resource requirement and consumption for nitrate ($NO_3^-$), ammonium ($NH_4^+$), or phosphorus (P). Colors indicate different species (see Table S1 for the full names of the species).

concentrations), which were not assessed in the monocultures (Fig. 3b; LMM: $F_{1, 10} = 3.999$, $P = 0.073$). In comparison, previous work has found that the classical Lotka-Volterra approach, which infers competition from the negative effects of one species on the growth of another, was sensitive to abiotic environments, such as water and nutrient levels[6–8]. Such context-dependency makes it challenging to predict from one condition to another (but see ref. 23 for a probabilistic approach). Furthermore, applying the Lotka-Volterra approach requires conducting experiments with $2^S$ -1 communities, where $S$ is the species richness (but see refs. 5,24. for approaches to reduce the experiment size). In contrast, the mechanistic approach sees the experiment size growing linearly with the species richness, as it requires measuring only monocultures. Since we found that the mechanistic approach also well-predicted community composition across species richness (Fig. 3c; mean accuracy > 74% for all species richness; Supplement Note 2) and time (Fig. S6), it is now poised to be applied to biodiverse communities under various conditions.

Although the predictive accuracy was high and did not differ between two-species, three-species, and four-species communities, we found that the accuracy was higher for two- communities than for six-species communities (Fig. 3c; $F_{1, 36} = 4.580$, $P = 0.039$). Alternative stable states[25] were present in communities and may explain this difference, particularly in more diverse communities (Supplement Note 6). Perturbations, such as those induced by refreshing the cultures every two days, could drive the community to transition between these states. Given the prevalence of disturbance and stochastic processes in natural communities, achieving accurate long-term predictions for species-rich communities likely requires continuous monitoring.

## The mechanistic rules are more likely to be met when competing for substitutable resources
We next used parameters that were quantified from the monoculture experiment and predicted whether the species pairs can stably coexist

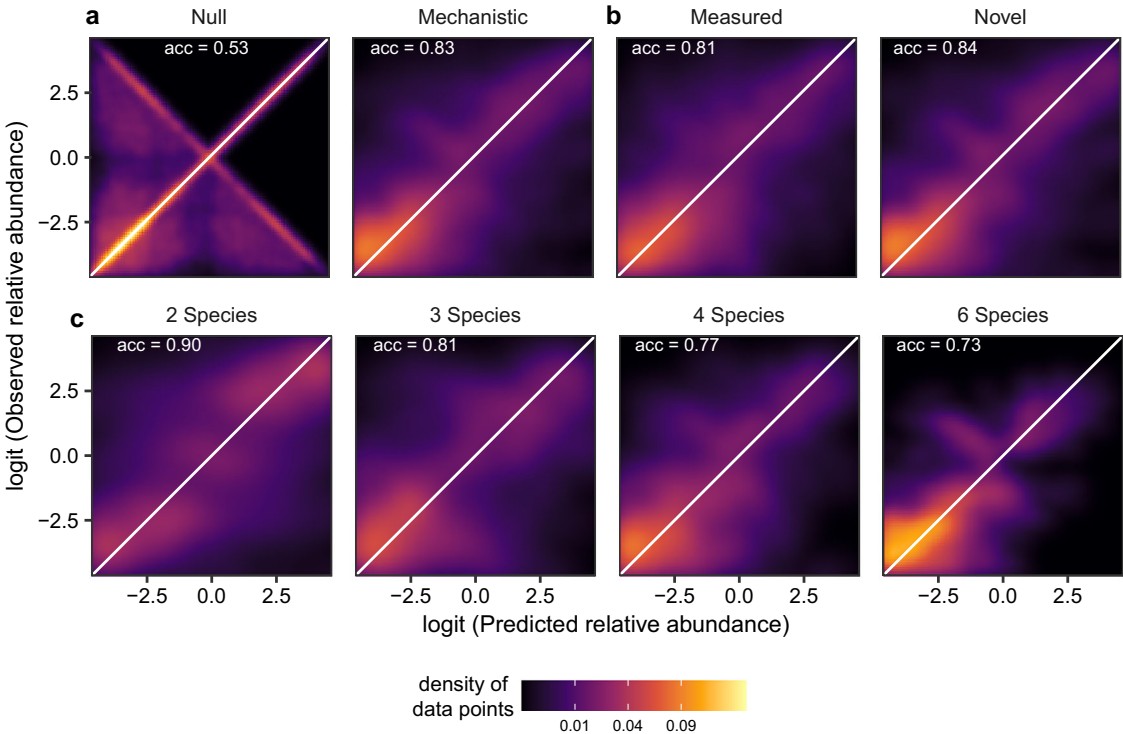

**Fig. 3 | The consumer-resource model predicts community composition. a** We predicted the composition of 960 communities with a null model that randomly shuffled species relative abundance within each community and a resource-consumer model that integrated resource requirement and consumption. The observed versus predicted relative abundances (logit transformed) are shown for each model, with warmer colors indicating a higher density of data points. Diagonal lines are included for reference and indicate perfect agreement between observations and predictions. The predictive accuracies (acc) are shown for each model. **b** The predictive accuracy did not differ between the measured (used in the monoculture experiment) and novel (not used in the monoculture experiment) conditions. **c** The predictive accuracy was higher for two-species communities than six-species communities.

by assessing Tilman's two rules on species coexistence. We found that when competing for two essential resources ($NO_3^-$ and P), only 30.3% of the species pairs (20 out of 66 all possible pairs) met Tilman's first rule for coexistence: each species must be limited by different resources (Fig. 4a). Among the species pairs that meet the first rule, only 40.0% of the pairs (8 out of 20) met the second: each species must consume more of the resource that more limits itself (Fig. 4b). Together, only 12.1% of the species pairs (8 out of 66) can stably coexist, which further depends on the resource supply[12]. When competing for substitutable resources ($NO_3^-$ and $NH_4^+$), the probability of stable coexistence was higher (22.7%, 15 out of 66), with 37.9% and 60.0% of the pairs meeting the first and the second rules, respectively (Fig. 4). Both findings indicate the low probability of stable coexistence when species compete for two limiting resources, regardless of the resource type. Previous studies[26,27] using the Lotka-Volterra approach revealed a comparable probability across taxonomic groups, ranging from 14.6% to 33.9%. However, because theory suggests multiple resources promote multispecies coexistence[28,29], future study should explore its effect.

To test whether these experimental results align with theoretical expectations for matching Tillman's rules, and whether there is a difference between competition for essential resources and competition for substitutable resources, we simulated 12 species over 999 times. For each species, we drew parameters for the consumer-resource model from unique distributions (Materials and Methods and Supplement Note 3.1). On average, 50% of the species pairs (66 pairs × 999 simulations) met Tilman's first rule (Fig. 3a; lower and upper 2.5% quantiles: 28.7% and 71.2%). Although the resource type did not affect the percentage of species that met the first rule, it did affect the second rule (Fig. 4b). Specifically, we found that species competing for essential resources were less likely to meet the second rule than those

competing for substitutable resources (average percentage 20.5% vs. 79.5%). Both findings confirmed the results of our experiments.

But why species that compete for essential resources are less likely to meet Tilman's second rule? Given the intrinsic link between consumption and growth rates (Fig. S8; Supplement Note 3.2), the resource that a species consumes more is often the same resource that supports higher growth rates, consequently more limiting the species (Supplement Note 4). However, this is only true for substitutable resource. For essential resources, Liebig's law of minimum reverses the results. Specifically, the only limiting resource is often consumed less and thus supports the lower growth (Supplement Note 4). This contrasts with Tilman's second rule: a species must consume more of the resource that more limits its growth. As a result, our finding, along with other theoretical work[30,31], explains the inherent challenges of species coexistence when competing for essential resources. However, substitutable resources for phytoplankton, such as different forms of nitrogen and different light spectra[32], is prevalent under natural conditions. Furthermore, phytoplankton serve as substitutable resources to their predators, offering an alternative pathway for the coexistence of phytoplankton[33] and for solving the paradox of plankton[34].

Last, we compared Shannon diversity of experiments and model simulations in multispecies communities. We used Shannon diversity rather than species richness to indicate the probability of coexistence in the experiment, because the duration of the experiment was not long enough to assess the endpoint of all communities (i.e., potential competitive exclusion). Shannon diversity at the end of the experiment increased with initial species richness (Fig. 5a; LMM: $F_{3, 36} = 7.64$, $P = 4.48 \times 10^{-4}$). This is especially the case when the species competed for substitutable resources, as indicated by the significant interaction between initial species richness and resource type (Fig. 5a; LMM: $F_{3, 575} = 22.88$, $P = 5.27 \times 10^{-14}$). Our simulation confirmed it: although no

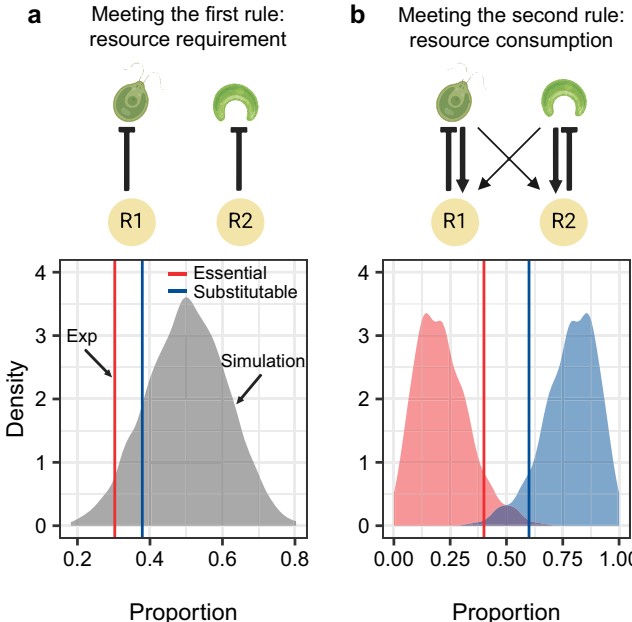

**Fig. 4 | The mechanistic rules are more likely to be met when two species compete for substitutable resources.** Vertical lines indicate the proportion calculated from the experiment (abbreviated as Exp) while density plots indicate the theoretical expectations calculated from 999 sets of simulation. **a** The simulation showed that the median proportion of two species meeting the first rule was 50%, irrespective of the resource type (essential or substitutable). The proportion found in the experiment was 30.3% and 37.9% (out of 66 species pairs) for essential and substitutable resources, respectively. **b** The proportion of two species meeting the second rule was low when competing for essential resources (40.0% in the experiment and 20.5% in the simulation) but was high for substitutable resources (60.0% in the experiment and 79.5% in the simulation).

more than two species can coexist when competing for two resources[35], the probability of coexistence of two species increased with initial species richness (Fig. 5b; Linear model (LM): $F_3 = 413.7$, $P < 2.2 \times 10^{-16}$), especially when competing for substitutable resources (LM: $F_3 = 21.54$, $P = 3.76 \times 10^{-16}$). This is because meeting the first rule is almost guaranteed in multispecies communities ($P = \frac{S-1}{S}$, where $S$ is the initial species richness; Supplement Note 4). Consequently, the probability of coexistence is predominately determined by the second rule, which is much more likely to be met for substitutable resources.

The effect of resource type on diversity was weaker in the experiment than the simulation, especially at low species richness. First, while the Shannon diversity did not significantly differ between species competing for essential resources and species competing for substitutable resources, coexistence under essential resources was potentially less stable (e.g., neutral[36]). Differentiating neutral from stable coexistence would require increased replicates to assess if communities consistently reach the same endpoints. Second, while our model assumes random parameters, introducing correlation between parameters (e.g., size-dependent consumption and growth rate; supplement Note 7) can reduce the effect of resource type. Third, while our model assumes that growth is directly linked to external nutrient concentration, real-world organisms can store nutrients, such as phosphorus. This storage capacity can influence species coexistence when competing for essential resources[37]. Last, because using $NO_3^-$ as nitrogen source is more energy-demanding than using $NH_4^+$, algae may preferentially use $NH_4^+$ when it is abundant. This makes the two resources not perfectly substitutable, thereby reducing the probability of coexistence. Exploring the full spectrum of resource complexity—from perfectly essential to perfectly substitutable—and its implications for coexistence could allow more nuanced predictions.

Our finding of the fundamental difference between essential and substitutable resources can advance our understanding of mechanisms governing species coexistence and evolution. For example, theoretical studies[38,39] revealed that competition for substitutable

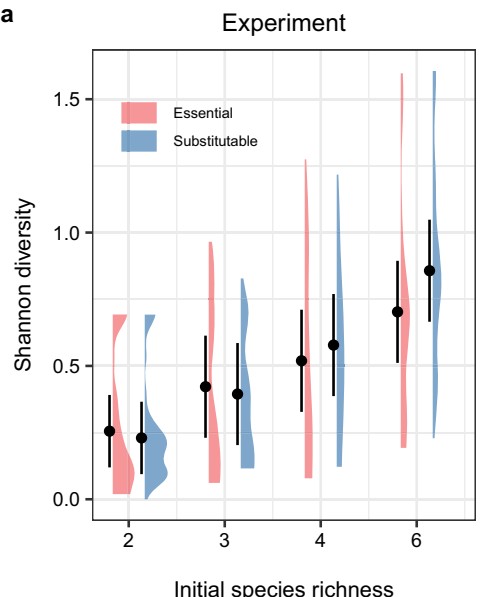

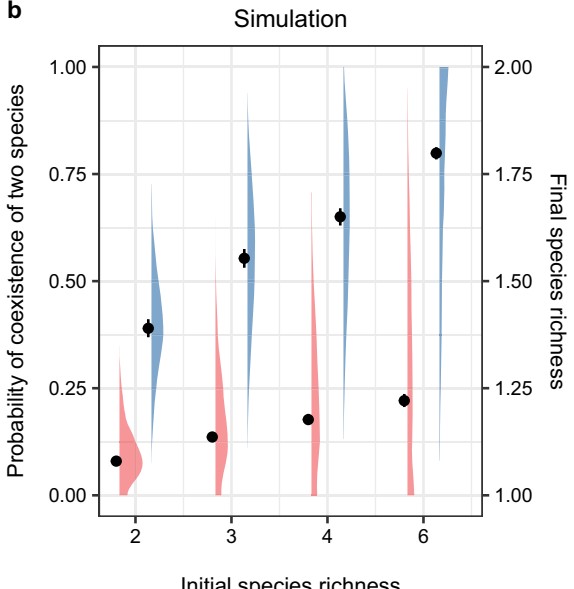

**Fig. 5 | Diversity is promoted in multispecies communities when species compete for substitutable resources.** **a** At the end of the experiment, Shannon diversity increased with initial species richness, especially when the species competed for substitutable resources. The sample sizes per resource type are 128 for two-species communities and 64 for other communities. **b** The 999 sets of simulation showed that the probability of coexistence of two species increased with

initial species richness, especially when competing for substitutable resources. Density plots indicate the distribution of Shannon diversity of all communities in the experiment and that of probability of coexistence of two species (i.e., meeting both mechanistic rules) calculated from the simulation. Error bars indicate the means and 95% confidence intervals.

resources led to character divergence[40], while competition for essential resources led to character convergence. As suggested by our study, this occurs because stable coexistence through essential resources is unlikely, making convergence toward ecologically identical species one pathway to achieving 'neutral' coexistence. Although we focused on resource competition in our phytoplankton system, our mechanistic model can be readily extended to other types of competition, such as cross-feeding in bacteria. Furthermore, our study suggests that the mechanistic model, which captures the processes underlying species interactions, can accurately predict community composition across a broad spectrum of abiotic and biotic conditions. This makes it a highly valuable tool for anticipating species loss and, ultimately, mitigating its pace.

## Methods

All the cultures were grown in 96-well plates (TPP®, Techno Plastic Products AG), at 20 °C, constant light (65 μmol/m²/s), and static conditions.

### Study species

We selected 12 green algal species (Table S1): *Acutodesmus obliquus* (abbreviation in Fig. 2: AO), *Chlamydomonas klinobasis* (CK), *Chlamydomonas oblonga (CO)*, *Chlamydomonas reinhardtii* (CR), *Chlorella vulgaris* (CV), *Chlorella minutissima* (ChlM), *Choricystis minor* (ChoM), *Monoraphidium griffithii* (MG), *Monoraphidium minutum* (MM), *Scenedesmus intermedius* (SI), *Scenedesmus quadricauda* (SQ), and *Scenedesmus sp* (SS). All species are from the algae collection at the Limnological Institute of the University of Konstanz. Before the experiments, stocks of each species were spread on agar plates and grown for seven days. Then, we picked a single colony for each species and grew it in modified WC medium (512 μM of nitrate and 25.6 μM of phosphorus) for nine days. The use of single colonies aims to minimize the effect of evolution on coexistence.

Two days before the experiments, we collected 2 mL culture for each species and washed the remaining nitrate and phosphorus by performing two rounds of centrifuging at 3000 G for 5 min, discarding approx. 1.9 mL supernatant and replacing it with modified WC medium that did not contain nitrate and phosphorus. After that, we counted each culture and diluted it to approx. 500 cells/μL with a modified WC medium that did not contain nitrate and phosphorus. We let the culture rest for two days to reduce the remaining nutrients in the medium or potential intracellular storage of nutrients.

### Quantifying resource use from monoculture experiments

On the day of the experiment, for each species, we added 10 μL of diluted culture (-5,000 cells) into 260 μL of modified WC medium. This medium was modified to contain varying concentrations of either nitrate resources ($NO_3^-$), ammonium ($NH_4^+$), or phosphorus. For $NO_3^-$, we reduced the nitrate concentration of the WC medium to one of the 12 levels: 0, 0.25, 0.5, 1, 2, 4, 8, 16, 32, 64, 128, and 512 μM. This ensured that nitrate was the major limiting factor, especially when low. For $NH_4^+$, we replaced sodium nitrate with ammonium chloride and employed the same 12 concentrations. For P, we reduced the P concentration to one of the 12 levels: 0, 0.0125, 0.025, 0.05, 0.1, 0.2, 0.4, 0.8, 1.6, 3.2, 6.4 and 25.6 μM. With two technical replicates, we had a total of 864 cultures (3 resources × 12 concentrations × 12 species × 2 replicates; a fully crossed design). All species and resource concentrations were randomized in the 96-well plates. We sampled each culture 20 μL after mixing for four consecutive days. Because our pilot experiment showed that most species experienced a lag phase of approx. 12 h, we started our first sampling 12 h after the inoculation. All culture preparation and sampling were done by a pipetting robot (OT-2, Opentrons). All samples were imaged and counted with a high-content microscopy (ImageXpress® Micro 4 High-Content Imaging System).

For a given species ($i$) and resource ($j$, $NO_3^-$, $NH_4^+$, or P), we fit the growth data with a consumer-resource model that contains one species and one resource:

$$\frac{1}{N_i}\frac{dN_i}{dt} = \frac{u_{\max,ij}R_j}{k_{ij}+R_j} - m_{ij} \quad (1)$$

$$\frac{1}{N_i}\frac{dR_j}{dt} = -\frac{c_{ij}R_j}{s_{ij}+R_j} \quad (2)$$

Equation (1) describes the per capita growth rate of species $i$, $\frac{1}{N_i}\frac{dN_i}{dt}$, as a function of the concentration of resource $j$ ($R_j$), and thus indicates resource requirement. It follows the Monod equation[41], where $u_{\max,ij}$ is the maximum birth rate of species $i$ on resource $j$, $k_{ij}$ is the half-saturation constant for birth rate (the concentration of resource $j$ when species $i$ reaches to half of its maximum birth rate), and $m_{ij}$ is the mortality rate of species $i$ in the presence of resource $j$. Equation (2) describes the per capita consumption rate of species $i$ on resource $j$, $\frac{1}{N_i}\frac{dR_j}{dt}$, as a function of $R_j$. It follows the type II functional response *sensu* Holling[42], where $c_{ij}$ is the maximum consumption rate, and $s_{ij}$ is the half-saturation constant for consumption rate (the concentration of resource $j$ when species $i$ reaches to half of its maximum consumption rate). We did not use type III functional response because it typically applies to biotic resources (prey), which can hide at low densities[43].

While the theoretical model is continuous, abundance can only be measured at discrete times. Consequently, we calculated $\frac{1}{N_i}\frac{dN_i}{dt}$ at time $t$ as $\ln\frac{N_{j,t+1}}{N_{j,t}}$, , and $\frac{1}{R_j}\frac{dR_j}{dt}$ at time $t$ as $\ln\frac{R_{j,t+1}}{R_{j,t}}$, where $t+1$ is one day after $t$. Theoretically, the relationship between resources and per capita growth rate (or consumption rate) can be linear. We tested this and found that the nonlinear relationship fitted the experimental data better, with a median R squared of 0.7 (Supplement Note 8).

We applied a Bayesian approach to parameterize the model with the *brms* (version 2.19.0) package[44] in R (version 4.2.3) language[45]. The resource consumption was quantified based on the combined information of per capita growth rate and initial resource concentrations (i.e., without the need for measuring resources over time). As a hypothetical example, consider a species growing in two media in which initial nitrate concentrations are 0 and 10 μM, respectively. While the initial per capita growth rate at the 10 μM concentration is high, it will decrease due to resource consumption and at a later time point reach the same rate as the initial per capita growth rate at the 0 μM concentration. Then, we can conclude that the amount of consumed nitrate is 10 μM. An additional assay in which we measured nitrate and phosphorus concentrations over time confirmed that our approach did well quantify the resource consumption (Supplement Note 9). To account for the non-independence of data from the same culture, we assumed that each culture had different maximum birth rates in the Bayesian model.

### Assessing community composition in competition experiments

In parallel with the monoculture experiment, we conducted competition experiments, where two, three, four, or six species were randomly chosen, mixed in the same ratio, and competed in modified WC media (20 μL culture added into 260 μL medium) for 12 days. We designed six resource conditions where the species competed for different ratios of essential resources ($NO_3^-$ and P) and another six where the species competed for different ratios of substitutable resources ($NO_3^-$ and $NH_4^+$; Fig. S2). Note that we maintained high concentrations of other resources (e.g., potassium) while keeping the manipulated resources relatively low. This ensured that the manipulated resources were the major limiting resources. Four of the twelve resource conditions were used in the monoculture experiments (i.e., measured conditions), and

the other eight were not (i.e., novel conditions; Fig. S2). For the species richness of two, we randomly selected 16 out of the 66 combinations of species. From these 16 combinations, we randomly selected eight combinations and then randomly added one, two, and four species, resulting in eight combinations each for species richness of three, four, and six, respectively. With two technical replicates, we had a total of 960 cultures (2 resource types × 6 resource conditions × 40 combinations × 2 replicates; a fully crossed design). We sampled 20 μL of culture after mixing on day one, and then every two days starting from day four. The algal abundance for each sample was counted with the high-content microscopy (ImageXpress®), and species were identified with a machine-learning approach (Supplement Note 10). To maintain semi-continuous cultures, starting from day four, we mixed 160 μL of the old culture and 80 μL of fresh medium every two days.

**Predicting community composition with the mechanistic model**
To assess whether the mechanistic model predicted the community composition, we started with a null model as a reference. In this null model, we randomly shuffled the species abundance in each community 999 times. Then, we used the mechanistic model to predict the community composition from day four to day twelve in the competition experiments. Specifically, we used the resource requirement and consumption that were quantified from the monoculture experiments (i.e., the mechanistic model). The initial abundance of each species is based on the abundance on day one of the competition experiments. We modeled the abundance of each algal species and resource with a consumer-resource model that contains multispecies and two resources:

$$\frac{1}{N_i}\frac{dN_i}{dt} = \begin{cases} \min\limits_{j=1,2}\left[\frac{u_{\max,ij}R_j}{k_{ij}+R_j} - m_{ij}\right] - D \; for \; essential \; resrouces \\ \sum\limits_{j=1}^{2}\left[\frac{u_{\max,ij}R_j}{k_{ij}+R_j} - m_{ij}\right] - D \; for \; substitutable \; resrouces \end{cases} \quad (3)$$

$$\frac{dR_j}{dt} = D(R_{j,in} - R_j) - \sum_{i=1}^{n}\frac{c_{ij}N_iR_j}{s_{ij}+R_j} \quad (4)$$

Equation (3) describes the per capita consumption rate of species $i$ when grown on two resources. D is the dilution rate. It equals to $\ln\frac{3}{2}$ day$^{-1}$ (160 μL old culture mixed with 80 μL fresh medium) at day 4, 6, 8, 10 and 12 and equals to 0 for the other days. All the other parameters follow the Eq. (1). For essential resources (here, $NO_3^-$ and P), the growth rate of the species is determined by the resource that supported a lower growth rate (i.e., Liebig's law). For a substitutable resource ($NO_3^-$ and $NH_4^+$), the growth rate is the sum of both growth rates. Our additional assay on growth rate under varying resource concentrations did support that $NO_3^-$ and P were essential resources and that $NO_3^-$ and $NH_4^+$ were substitutable resource (SI text 10). The Eq. (4) describes the change of resource $j$, as a function of $R_j$ and $N_i$. $D(R_{j,in} - R_j)$ describes the resources supply, with $R_{j,in}$ representing input nutrient concentration (same as the initial nutrient concentration). $\sum_{i=1}^{n}\frac{c_{ij}N_iR_j}{s_{ij}+R_j}$ describes the resource consumption by all the species, with $n$ representing the species richness in the community (i.e., $n = 2, 3, 4,$ or $6$). All the other parameters follow Eq. (2). Note that, for simplification, we transposed $R_j$ from the left to the right side in Eq. (4). This resulted in the difference in the left-hand sides of Eqs. (2) and (4) but did not affect the prediction.

To assess the performance of our models in predicting community composition of competition experiments, we calculated predictive accuracy using the Bray–Curtis similarity index[46] for each community at each time point (Eq. 5):

$$acc = \frac{2\sum_{i=1}^{n}\min(a_{i,obs}, a_{i,pred})}{\sum_{i=1}^{n}(a_{i,obs} + a_{i,pred})} = \sum_{i=1}^{n}\min(a_{i,obs}, a_{i,pred}) \quad (5)$$

where $n$, $a_{i,obs}$, $a_{i,pred}$ are the number of species in the community, observed and predicted relative abundance of species $i$, respectively. The value of accuracy ranges from 0 to 1. A higher value indicates higher predictive accuracy, with a value of 1 indicating that the predicted relative abundance is exactly the same as the observed relative abundance. To minimize the effect of randomness on cell sampling and counting, we excluded data points where fewer than 10 cells were counted. We also calculated the predictive accuracy according to the absolute abundance, the results were overall similar (Supplement Note 2).

We first tested whether the mechanistic model better predicted the community composition than the null model. To do so, we conducted a linear mixed model with the *lme4* (version 1.1-33) package[47]. The model included the predictive accuracy as the response variable and type of prediction (the null vs. the mechanistic model) as the fixed effect. We then tested whether the predictive accuracy was consistent across time point, across species richness, and across measured and novel environments (e.g., if a species fails to grow in a given medium in monoculture, the model will naturally and accurately predict it to be outcompeted in this measured medium when competing with other species). To do so, we conducted another linear mixed model. The model included the predictive accuracy of the mechanistic model as the response variable and the time point (day four to day twelve), the type of medium (novel *vs.* measured), and species richness as the fixed effect. We did not include the interactions between the main effects because this is not the main interest of our study and because this will lead to many categories (e.g., 5 time points × 4 species richness × 2 types of medium). Both models included the identity of medium and species combination as the random effects. The predictive accuracy was logit-transformed to improve the normality of the residuals. For species richness and time points, which contain more than two levels, we created dummy variables[48] to compare each levels to the reference levels (i.e., two species, and day 4, respectively). The significance of the fixed effects was assessed with ANOVA.

**Assessing the proportion of species that meet Tilman's rules for coexistence**
**Community of two consumers.** To assess the proportion of species pairs that can stably coexist, we assessed Tilman's two rules on species coexistence, using parameters that were quantified from the monoculture experiment. To assess the proportion of species pairs that met Tilman's first rule, each species must be limited by different resources, we tested whether the zero net growth isoclines (ZNGIs) of two species have a positive intersection (i.e., resource equilibrium) for each species pair (Supplement Note 4). This was done with the *sympy* (version 1.11.1) library[49] in python (version 3.10.9)[50].

For species pairs that met the first rule, we further assessed the proportion of species pairs that met Tilman's second rule, each species must consume more of the resource that more limits itself. To do so, we determined for each species the resource that more limited itself. Mathematically, this is done by comparing the partial derivatives of the per capita growth rate of each species to each resource at the equilibrium, using the *math* library[51]. Then, we determined for each species the resource that it consumed more by comparing the per capita consumption rate of each species on each resource at the equilibrium.

To test whether our experimental results align with the theoretical expectation, we simulated 12 species. For each species, we randomly drew the parameters (e.g., $c_{ij}$ and $s_{ij}$) in the consumer-resource model from zero to one following a uniform distribution. Note that changing the range of the distribution (e.g., using from zero to two) will not affect the results (Supplement Note 4). Changing the type of the distribution (e.g., using Gaussian distribution) will affect the results quantitatively but not qualitatively (Supplement Note 4). We set the mortality rate to 0.1 for all the species. This

is because our experiments showed that the species-specific mortality rate (mean: 0.01 day$^{-1}$) is much lower than the mortality caused by dilution, which is constant.

Instead of modeling the per capita growth rate as a function of resource concentration, as we did earlier to quantify resource requirement, we modeled it as a function of per capita consumption rate. While the former approach is more practical for parameter estimation, the latter is theoretically justified, as an individual converts its consumed resources into growth[38] (but see Supplement Note 3 for how the two methods match with each other). Specifically, we modeled the per capita growth rate of species $i$ as:

$$\frac{1}{N_i}\frac{dN_i}{dt} = \begin{cases} \min_{j=1,2}\left[\frac{w_{ij}\,g_{ij}(R_j)}{q_{ij}+g_{ij}(R_j)}\right] - m \; for \; essential \; resrouces \\ \sum_{j=1}^{2}\left[\frac{w_{ij}\,g_{ij}(R_j)}{q_{ij}+g_{ij}(R_j)}\right] - m \; for \; substitutable \; resrouces \end{cases} \quad (6)$$

where $w_{i,j}$ is a weighing factor, the value of one unit of the consumed resource $j$ to species $i$. $g_{ij}(R_j)$ is the per capita consumption rate of species $i$ on resource $j$, i.e., $\frac{c_{ij}N_iR_j}{s_{ij}+R_j}$. $q_{ij}$ is the half-saturation constant for birth rate (the level of consumed resource $j$ when species $i$ reaches half of its maximum birth rate). $m$ is the mortality rate. For the changes of resource, we modeled it as Eq. (4).

We only kept species that can grow when alone (i.e., had positive abundance when reaching equilibrium), which meets the condition of our experiment. As above, we assessed the proportion of species pairs that met Tilman's rules. We repeated the simulation over 999 times and calculated the average and 95% quantile for both proportions. Besides the simulation, we also derived the analytic solution in the linear system (Supplement Note 4), which confirmed the results of our simulation.

Note that as long as the two rules are met, the coexistence is further determined by the resource supply: the resource supply must fall within the region bounded by the consumption vectors of the two species (the third rule). This is explored in the Supplement Note 4.4).

**Community of multiple consumers.** While Tilman's mechanistic rules are based on two consumers, we scaled it up to multiple consumers (Supplement Note 5). The first rule should be adapted to: at least two of the species must be limited by different resources. Additionally, the equilibrium of these two species must not be invadable by other species (i.e., all the other species have a negative growth rate at this equilibrium). Otherwise, at least one of the two species will be replaced by a third species. The second rule stays the same: for these two species, each species must consume more of the resource that more limits itself.

We calculated the mean probability of coexistence for each of the 999 sets of simulations above. For each simulation, we used all the combinations of two species (66), and randomly selected 99 combinations of three, four, or six species. Then, we tested whether the mean probability of coexistence of two species depended on the resource type and initial species richness with a linear model. The model included the mean probability of coexistence of each simulation as the response variable and the resource type, initial species richness, and their interaction as the explanatory variables.

In comparison, we tested with our competition experiment whether the coexistence depended on the resource type and initial species richness with a linear-mixed effects model. We excluded data from conditions where one of the two manipulated resources was abundant (e.g., 0.1 μM P and 512 μM NO$_3^-$), as these scenarios were limited by a single resource and could not be categorized as either essential or substitutable. The model included the Shannon diversity at the end of the experiment as the response variable; the resource type, initial species richness, and their interaction as the fixed effects; and the

species combination (as we have two technical replicates) and resource condition as the random effects. Unlike the simulation, we used Shannon diversity. This is due to the duration of the competition experiment being too short to assess the endpoint of co-existence. However, by investigating Shannon diversity, we can still glean insights into whether the initial species richness and/or resource type promote coexistence.

### Reporting summary
Further information on research design is available in the Nature Portfolio Reporting Summary linked to this article.

## Data availability
The data generated in the study have been deposited in the Figshare database (https://doi.org/10.6084/m9.figshare.29093678).

## Code availability
The code of the study has been deposited in the Figshare database (https://doi.org/10.6084/m9.figshare.29093678).

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

## Acknowledgements

We thank Chuliang Song for help with deriving the analytic solution, Natascha Handke for helping with the experiment, and Faruk Tök for conducting a pilot experiment. We also thank Toni Klauschies, Ville Mustonen, and Dietmar Straile for comments on the manuscript. Z.Z. was supported by Deutsche Forschungsgemeinschaft (DFG: ZH 113/2-1) and YSF co-funding at the University of Konstanz.

## Author contributions

Z.Z. and L.B. conceived the idea and designed the study. Z.Z. performed the experiment, analyzed the data, and conducted the simulation. Z.Z. drafted the manuscript and L.B. revised it.

## Funding

## Competing interests

The authors declare no competing interests.
