## [Transparent Peer Review file · Nature Communications]

Mechanistic prediction of community composition across resource conditions and species richness

Corresponding Author: Dr Zhijie Zhang

Version 0:

Reviewer comments:

Reviewer #1

(Remarks to the Author)

Understanding species coexistence has been a long-lasting quest in Ecology for decades. In the 80s where Tilman proposed the mechanistic rules of species coexistence based on the species differences in the use of limiting resources but the critical requirements have not been fully empirically explored. Zhang and Becks provided an empirical test for these mechanistic rules. Using the combination of analytical simulation and microbiological experiments, they show that the second rule is more likely to be met when two species competed for substitutable resources and the resource that a species consumes more supports the higher growth (more limiting). The experiment is novel and the results are clear. I recommend the manuscript acceptance in Nat Comms. pending the authors addressing my comments below.

Major comments

- The experimental design has to be better explained. Throughout the methods, it's unclear to me which resources combination were chosen and tested with which species combinations. It'd be helpful to see a full illustration of the setup in figure 1.
- Are all species able to grow in monoculture in the resource environment where they were tested in the community? For example, if a species is used in a community under a P limiting condition and it also cannot grow on its own in the same P limiting condition, this may inflate the accuracy. Because I'd expect in either monoculture or community (2, 3, 4, 6, whatsoever), this species always does not grow, assuming cross-feeding does not happen in the plankton system.
- line 131 "stable coexistence". If I understand correctly, each species combination was performed at one initial relative abundance (ie, all species have equal cell count at the beginning of the experiment)? Can the authors comment on how this can be experimentally distinguished from neutral coexistence?
- I am curious of the competitive ranking among these species. Based on fig1ab, CR is the most competitive species with regard to each nutrient use ability. Heuristically, is the more nutrient efficient species in monoculture more likely to be a dominant species in the community?
- The main analysis that supports the authors' statement of "predicting coexistence across environments" is the measured vs. novel conditions in figure 2. While the mechanistic approach provided by the authors is merely the first step, I would argue that resource complexity beyond substitutability could largely alter the results. For example, metabolically speaking, ammonium and nitrate are very similar, but not entirely substitutable. They may differ in the energy costs as nitrate is one step away from amino acids than ammonium. Can the authors discuss the environmental contexts with more resource complexity (i.e., more diversity type of resources, or simply more resources of the same type)?

Minor comments

- It seems that "960 communities" result from 40 species combinations * 2 replicates * 12 resources concentrations. Based on the abstract, does the total experiment you did sum up to 960? Or did you do 960 for nitrate, 960 for ammonium, 960 for P, 960 for nitrate+ammonium, and 960 for P + nitrate? Figure S1 is helpful and I would like to see a similar figure for resources too.
- I think the notation -N after the two N resources is redundant and it impedes the readability
- Does the result hold when the parameters are fitted with models considering a type III functional response instead of type I (linear) and type II (monod)?
- Can you show the raw relative abundance data and the richness change for your communities? For example a barplot of relative abundance of start vs final

Intro

- line 58: Do the monocultures experience two limiting resources simultaneously (for instance, both low P and low N), or only one limiting resource at a time (low P and excessive N, or excessive P and low N)? I know the resource combination are shown in Fig S2, but it'd help the reader to know how the resource combinations were set up (are the). Is it 12 concentrations of each of the limiting resource (it looks like it in Fig1b). 12 species * 12 resources concentrations * 3 resources = 432? Based on Fig S2, it seems that it's $12 \times 2^3 = 312$?

Results

- line 75: It reads like experimental data but the figures shows the fitted parameters. Could you also provide the raw data (e.g, per capita growth for at a nutrient frequency) for each species? It can be either overlaying on figure 2ab or in a separate supplement figure.

- line 78: how is the composition of the 960 communities set up? See my major comment above.

- line 79: what's null? Randomly shuffle the species frequency within a to-be-predicted community?

- I would recommend using relative abundance instead of species frequency since it can get confused with accuracy ["frequency"].

- line 81: What's the R^2 and RMSE of the linear model if you were to predict each species' accuracy?

- line 98: what's a novel condition?

- line 108: in fig2e, the mean accuracy of 4-sp and 6-sp communities do not look like > 80%

- Is it also the case for each species richness X day interaction?

- line 125: how is the proportion of species pairs calculated?

- line 133: how is the stable coexistence defined?

- line 140: do you simulate all choose(12,2)=66 pairs?

Methods

- line 192: what's TPP?

- line 203: use μM instead of uM

- line 222: Not sure I fully understand why there are 864 cultures? Isn't it 3 resources * 12 concentrations each * 2 replicate * 40 species combinations = 2880?

- line 273: it's unclear to me which species combination and resource combination were chosen in the competition experiments. Fig S2 does not show the which are the novel conditions. Is it all. It will be a long list but could you provide a list of resourceXspecies combinations you tested?

- line 274: Does it mean you randomly selected 8 out of 66?

Fig1

- panel d: Again, explain the novel conditions.

- panel e: Do the prediction accuracies of 3-sp and 4-sp communities differ from the 2-sp ones?

- What's the sample sizes of 2, 3, 4, 6-sp communities here for each density plot?

Fig2

- panel c-e. What does the "[frequency]" mean in the y axis? Does it mean the accuracy is computed based on BC similarity of species frequency? It's confusing.

- panel d: Does a novel condition mean that the resource combination where the predicted community was assembled is not used in monoculture experiment at all? I would include these terms in figure 1 to improve clarity.

Supplements

- line 13: typo "affected". Also, what are the 4 combinations that are not significantly affected? Are they all pairs, 6-species community or what? Based on functional redundancy, I would expect that those communities that are not affected by resource conditions are more likely to be initially diverse communities rather than pairs

- line 50: I would specify both c and s in this MM equation as these are two randomly drawn parameters later in the simulation

- line 186: typo. 0.8 for substitutable resources

- line 323: could you provide more details on how these cell features were measured? It'd be very helpful for people working on the the same system.

- Fig S2: black dots instead of red dots?

- Fig S3. Not sure I understand the rationale behind PCA of beta-distance matrix. Why don't you used PCA directly on the relative abundance at 12day? Also, It'd be nice to see the final community richness/relative abundance for each culture in a bar chart. I wonder if the U-shape commonly seen in these PCA plots is the artifact of low diversity at the end of the experiments (at least it should be the case for combinations 1-16 where the initial communities are pairs)

(Remarks on code availability)

The scripts are not available for review.

Reviewer #2

(Remarks to the Author)

Recently, many experiments have shown that resource competition is essential in the assembly of diverse microbial communities in complex environments by reproducing experimental observed patterns with consumer-resources models (Elife 11 (2022): e75168., Nature Microbiology 9.4 (2024): 1036-1048). Given the complexity of these interactions, coarse-graining approaches become necessary, as many models contain numerous underdetermined parameters.

In this study, the authors focus on a simpler, controlled system—monoculture experiments with green algae. They determine parameters for consumer-resource models (CRM) from experimental data and directly evaluate Tilman's rules of competition from the models, offering a fresh perspective on resource substitutability and its role in promoting diversity.

Here are my major comments:

1. The authors emphasize the well-known limitations of Lotka-Volterra (LV) models. However, they could strengthen their argument by focusing more on the experimental system used, especially how the plankton ecosystems compare with bacteria ones. Because of cross-feeding, microbial systems tend to have complex interactions, resulting in difficulties in precise modelling. Are cross-feeding effects important in green algae? If resource competition indeed dominates, this system could provide a more ideal framework for CRM theory validation. It would be nice to have some discussions on this point.
2. Line 39, the authors used "phenomenological approach". It seems they may have adopted this terminology from Tilman's original paper, but the term 'phenomenological' may be misleading. In this paper, the authors used a reasonable model (CRM instead of LV) but the approach is still "phenomenological" as it still uses models to explain experimental phenomenon. I recommend deleting the term "phenomenological" for clarity and precision.
3. The manuscript lacks direct comparisons between experimental data and model simulations, which weakens the overall argument.
 - 3.1. In Fig. 1c, 2a and others, the absence of direct comparisons between experimental data and model outputs makes it difficult to assess the accuracy of the inferred parameters. Including the original data points in these figures would significantly improve their interpretability and allow for a more thorough assessment of model fit.
 - 3.2. In Fig 2 c-e, the authors use Bray-Curtis similarity as the metric for accuracy. However, given the small number of species involved, this may not be the most appropriate choice. A direct comparison of community composition between simulations and experimental results could offer more insight into model performance.
 - 3.3. In Fig 3, While I understand the intent behind Figure 3, the two vertical lines drawn on the probability density may confuse readers as they represent distinct aspects. I suggest moving some theoretical results from the SI to the main text and then using the probability density to reinforce those theoretical conclusions would be more helpful.
 - 3.4. In Fig. 4, the distinction is notable only at Richness=6. I suggest the authors further explain this inconsistency with theoretical predictions. One possible explanation is that the real feasible region for coexistence is not as large as the angle it spans due to undersampling issues. Alternatively, the authors may wish to discuss if it results from the limitations of the experiments, as mentioned in the Methods section. A deeper analysis and explanation of this inconsistency would enhance the manuscript.

While the conclusions and theoretical analysis presented in this paper are promising, the manuscript focuses primarily on predictions from consumer-resource models in simple ecosystems. As such, it is crucial to achieve stronger self-consistency and alignment between the models and experimental results, particularly in comparison to similar studies on more complex environments. Unfortunately, the current manuscript lacks sufficient direct comparisons between experiments and simulations to validate the robustness of the models. Furthermore, some of the conclusions drawn appear weakly supported by the available data. Therefore, I am unable to recommend this manuscript for publication in its present form.

Minor comment:

Caption of Figure S2 in SI: BLACK dots indicate the competition experiments. Or you may redraw the figure with red dots to be consistent with Figure S18.

(Remarks on code availability)

Reviewer #3

(Remarks to the Author)

MacArthur's consumer-resource model (CRM) has become a staple of modern ecological theory, serving as one of the primary vehicles for building intuition about the factors that affect species coexistence. It has the great merit of being the simplest model that respects the most basic physical constraint on population dynamics: a population can expand only by converting materials from its environment into biomass. The CRM achieves its simplicity by assuming that the availability of these resources is the dominant factor in determining population growth rates.

Despite the importance of the CRM for theoretical ecology, it has proven remarkably challenging to explicitly test experimentally. Ideally, one would begin by testing the model on microbial populations, where the experimental conditions can be precisely controlled, and the spatial and temporal scales allow for many replicates in a study of reasonable duration. In these systems, however, the resources are generally small molecules, whose abundances are barely detectable in the regime of resource-limitation where the CRM is most likely to be valid.

In this manuscript, the authors seek to begin addressing this lacuna, by testing a generic and ecologically meaningful prediction of the CRM on a large set of combinations of species of algae. Specifically, they focus on the stability conditions for the two-species fixed point of CRM dynamics: under a biologically plausible parameterization, these conditions are usually satisfied in competition for substitutable resources, while they are harder to guarantee in the case of essential

resources. The authors quantify this theoretical claim using a specific implementation of the CRM with randomly sampled parameters, showing that the probability of stability is indeed much lower for essential resources.

To test this prediction, the authors estimate the CRM parameters from monoculture experiments with known initial resource concentrations. They face the same obstacle as prior efforts to test the CRM, and are unable to detect the resources when their concentrations approach the level where they begin to limit growth. But they circumvent this difficulty by performing a global fit of the model to all the monoculture experiments performed with the same species in various starting concentrations of the same resource. The variety of initial conditions compensates for the absence of full time-series data for the resources.

With such a well-posed and relevant question and a sound experimental method, this manuscript should eventually be published in a journal like Nature Communications. In my view, however, the results deserve further analysis before the article is released. The immediate message of Figures 3 and 4 is much more interesting than the authors' discussion would indicate, since the data clearly contradict the theoretical expectations. Whereas stable coexistence was supposed to be much more difficult for essential resources, the inferred parameters satisfy the stability conditions in about half the combinations for both kinds of resources, and the distribution of final diversity in the multi-species experiments in the two cases is almost indistinguishable. For me, this unexpected result is the principal message of the article, opening up promising possibilities for further investigation. To finish the project, the authors should at least propose one or two plausible hypotheses to explain the contradiction, possibly involving a few additional experiments to determine which is most viable.

My own hunch is that the problem lies in the assumption of independent random parameters in the probability calculations. At the most basic level, the function that maps resource concentration to population growth should include a scaling factor that depends on the species' cell size, as the nutrient uptake rate scales with cell surface area while the resource requirements scale with volume. This will introduce correlations among the parameters for a given species that will alter the probability calculus. If this is insufficient to explain the data, it may be worth investigating the biological factors that determine net uptake stoichiometry in more detail, to see whether physical constraints perhaps force the impact vectors to be sufficiently similar as to admit a metastable equilibrium.

Specific comments:

1. Figure 1a is confusing, especially the top panel where it seems like the "superior" species is not limited by any resource, but I don't see any obvious way of improving it. Since the paper is about Tilman's rules, one could simply begin immediately with ZNGI's and impact vectors. This may be more difficult for some readers to follow, but would be useful for those most interested in the result.
2. In Figure 2, I would like to see at least one example of the measured per-capita growth rate vs. (inferred) resource concentration as compared with the Monod fit, and not be forced to re-plot the data myself from the repository in order to get a feel for what the actual measurements look like. Perhaps the authors could select two or three representative species and plot the measurements together with the fits, and save the rest of the curves for the supplement.
3. The model without resource consumption (as in Figure 2c) seems hard to justify as a meaningful point of comparison, since in this model all the population densities are forever expanding exponentially, quickly reaching physically impossible levels. This strangeness is hidden by the use of relative species frequencies in place of absolute population sizes, but that does not change the unphysical character of the model. The authors should eliminate this comparison, or devise another way of testing the relevance of the impact vector.
4. On page 8, line 159, the authors relate their results to the paradox of the plankton. Most discussions of the paradox of the plankton focus on Tilman's first rule, noting that the intersection of all the ZNGI's demands fine-tuning of parameters when the number of species exceeds the largest plausible resource dimension. Since this article concerns the second rule, the reference requires more explanation, or should be eliminated.

(Remarks on code availability)

Version 1:

Reviewer comments:

Reviewer #1

(Remarks to the Author)

The authors have addressed all the comments I raised and greatly improved the clarity of the manuscript. I am generally satisfied with the revisions made. Here are some minor comments for further consideration:

- Methods and Supplements. Please use mL instead of ml.
- Lines 232 and 234. Please remove -N after nitrate and ammonium.
- Line 238. "All species and resource concentrations were randomized." I thought it's a fully crossed design? Or are they randomized on the 96-well plate?
- Line 253. Is mortality rate really dependent on the resource? If so I think it should mean mortality in the presence of resource. If not, it should be just m_i .
- Fig2 and S2. I recommend moving the point legend outside of the panel. It's somewhat confusing to have open legends within the panel.
- FigS3 caption. Typo modedlled.

- Fig S6. Regarding the sample sizes for 2, 3, 4, 6-species communities, I think it's 384, 192, 192, 192 which adds up to a total of 960.

(Remarks on code availability)

Yes, a README file and main data table are included. The Rmd and python scripts are self-describing. I would appreciate including the scripts for the image processing pipeline as it will hugely benefit researchers working on similar platforms.

Reviewer #2

(Remarks to the Author)

My concerns have been well addressed.

(Remarks on code availability)

Reviewer #3

(Remarks to the Author)

The authors have taken significant steps to address all my concerns from the first report. I do recommend including the cell-size scaling factor analysis in the current study, at least in the supplementary material, with a mention in the main text. The fact that the rule 1 probability drops to 40% -- very close to the substitutable-resource experimental value of 38% -- removes the open question about the source of the hidden correlations among parameters in the experiment, and helps strengthen the main result.

It would also be useful to compute how the probabilities change for rule 2. The authors claim that these probabilities are unaffected by the scaling factor, but they neglect the fact that these ratios still depend on the equilibrium resource concentrations, which are affected.

(Remarks on code availability)

REVIEWER COMMENTS

Reviewer #1 (Remarks to the Author):

Understanding species coexistence has been a long-lasting quest in Ecology for decades. In the 80s where Tilman proposed the mechanistic rules of species coexistence based on the species differences in the use of limiting resources but the critical requirements have not been fully empirically explored. Zhang and Becks provided an empirical test for these mechanistic rules. Using the combination of analytical simulation and microbiological experiments, they show that the second rule is more likely to be met when two species competed for substitutable resources and the resource that a species consumes more supports the higher growth (more limiting). The experiment is novel and the results are clear. I recommend the manuscript acceptance in Nat Comms. pending the authors addressing my comments below.

Major comments

- The experimental design has to be better explained. Throughout the methods, it's unclear to me which resources combination were chosen and tested with which species combinations. It'd be helpful to see a full illustration of the setup in figure 1.

Response: We have illustrated the resource combinations in Fig.1 (details in Fig. S2) and made it clearer that the experiment used a fully crossed design (L238).

- Are all species able to grow in monoculture in the resource environment where they were tested in the community? For example, if a species is used in a community under a P limiting condition and it also cannot grow on its own in the same P limiting condition, this may inflate the accuracy. Because I'd expect in either monoculture or community (2, 3, 4, 6, whatsoever), this species always does not grow, assuming cross-feeding does not happen in the plankton system.

Response: All species were able to grow under limiting resources in monocultures. We have mentioned this now (L76). *Monoraphidium griffithii* did not grow well when P concentration was low or when nitrate concentration was low (i.e., only ammonium as the only or main N source). However, removing communities that contained *M. griffithii* did not affect the overall predictive accuracy. Consequently, we believe that our results are robust.

Cross-feeding between phytoplankton species might be very rare given that very few (if any) studies have reported this. We have now introduced and discussed cross-feeding (L52-57 & 199-201).

- line 131 "stable coexistence". If I understand correctly, each species combination was performed at one initial relative abundance (ie, all species have equal cell count at the beginning of the experiment)? Can the authors comment on how this can be experimentally distinguished from neutral coexistence?

Response: Yes, each species combination had one initial relative abundance. We did not experimentally test whether the coexistence was stable or not, which requires much longer time. Instead, we predicted with the parameters, whether the species can coexist or not. We have made it clearer now (L122-124).

To experimentally test whether the coexistence is stable or neutral, one need to increase the number of replicates and to determine whether they always reach the same endpoint. We have discussed it now (L183-185).

- I am curious of the competitive ranking among these species. Based on fig1ab, CR is the most competitive species with regard to each nutrient use ability. Heuristically, is the more nutrient efficient species in monoculture more likely to be a dominant species in the community?

Response: If we understand correctly, the reviewer meant whether the species with the highest maximum growth rate (CR) was more likely to be a dominant species. We found no significant correlation between maximum growth rate and final relative abundance. Please let us know if we have misunderstood the question.

- The main analysis that supports the authors' statement of "predicting coexistence across environments" is the measured vs. novel conditions in figure 2. While the mechanistic approach provided by the authors is merely the first step, I would argue that resource complexity beyond substitutability could largely alter the results. For example, metabolically speaking, ammonium and nitrate are very similar, but not entirely substitutable. They may differ in the energy costs as nitrate is one step away from than ammonium. Can the authors discuss the environmental contexts with more resource complexity (i.e., more diversity type of resources, or simply more resources of the same type)?

Response: This is an interesting point, which we have now discussed (L188-192).

Minor comments

- It seems that "960 communities" result from 40 species combinations * 2 replicates * 12 resources concentrations. Base on the abstract, does the total experiment you did sum up to 960? Or did you do 960 for nitrate, 960 for ammonium, 960 for P, 960 for nitrate+ammonium, and 960 for P + nitrate? Figure S1 is helpful and I would like to see a similar figure for resources too.

Response: It is 40 species combinations * 2 replicates * 12 resources combinations. We have clarified this in the main text (L294-295) and added figures on the resource environments (Fig. 1b & Fig. S2).

- I think the notation -N after the two N resources is redundant and it impedes the readability

Response: We have removed the "-N".

- Does the result hold when the parameters are fitted with models considering a type III functional response instead of type I (linear) and type II (monod)?

Response: We initially considered type III functional response in the very early stages of the study. However, the model fitting was poor, likely due to the complexity of the function (e.g., the inclusion of a quadratic term for resource density). In biological systems, type III response typically occurs when predators need time to learn how to capture prey or when prey can hide at low densities. Both scenarios are less applicable to abiotic resources (here N and P) than biotic prey. We have mentioned this now (L258-259).

- Can you show the raw relative abundance data and the richness change for your communities? For example a barplot of relative abundance of start vs final

Response: We have presented the change of relative abundance now (Fig. S5). The species richness did not change in most of the communities. There are two reasons behind. First, the experiment is not long enough to observe competitive exclusion in all communities. Second, the machine learning model, while classifying species with high accuracy, could classify some cells as an already excluded species, thus overestimating the final richness.

Consequently, we presented the Shannon diversity instead (Fig. 5) and did not show the richness change.

Intro

- line 58: Do the monocultures experience two limiting resources simultaneously (for instance, both low P and low N), or only one limiting resource at a time (low P and excessive N, or excessive P and low N)? I know the resource combination are shown in Fig S2, but it'd help the reader to know how the resource combinations were set up (are the). Is it 12 concentrations of each of the limiting resource (it looks like it in Fig1b). 12 species * 12 resources concentrations * 3 resources = 432? Based on Fig S2, it seems that it's $12 * 23 * 2 = 312$?

Response: The monocultures only experienced one limiting resource at a time. Consequently, we have 432 monocultures (12 species * 12 concentrations * 3 resources * 2 replicates). We have made it clearer (L238) and added more information in the legend of Fig. S2.

Results

- line 75: It reads like experimental data but the figures shows the fitted parameters. Could you also provide the raw data (e.g, per capita growth for at a nutrient frequency) for each species? It can be either overlaying on figure 2ab or in a separate supplement figure.

Response: We have now provided an example in Fig. 2a and all species per resource in Fig. S3.

- line 78: how is the composition of the 960 communities set up? See my major comment above.

Response: see our response above.

- line 79: what's null? Randomly shuffle the species frequency within a to-be-predicted community?

Done (L81-82)

- I would recommend using relative abundance instead of species frequency since it can get confused with accuracy ["frequency"].

Done.

- line 81: What's the R^2 and RMSE of the linear model if you were to predict each species' accuracy?

Response: We have added the R^2 and RMSE in Table S2.

- line 98: what's a novel condition?

Response: It is the conditions that were not used in the monoculture experiment. We have clarified it (L98).

- line 108: in fig2e, the mean accuracy of 4-sp and 6-sp communities do not look like $> 80\%$

Response: No, the mean accuracy of 4-sp and 6-sp was below 80%. We have revised it (L108) and added the accuracy for each species richness in Fig. 3.

- Is it also the case for each species richness X day interaction?

Response: Although the species richness x day interaction was significant, we did not specifically test the species richness X day interaction. This is because there are too many levels to interpret and because this is not the main interest of the study. We have mentioned this now (L349-352).

- line 125: how is the proportion of species pairs calculated?

Response: It the number of pairs meeting the rule divided by the number of all possible pairs (66). We have added this (L125).

- line 133: how is the stable coexistence defined?

Please see our response above.

- line 140: do you simulate all $\text{choose}(12,2)=66$ pairs?

Response: Yes, we have added this now (L143).

Methods

- line 192: what's TPP?

Response: It is the abbreviation of the company name. We have revised it.

- line 203: use μM instead of uM

Done.

- line 222: Not sure I fully understand why there are 864 cultures? Isn't it 3 resources * 12 concentrations each * 2 replicate * 40 species combinations = 2880?

Response: See our response above.

- line 273: it's is unclear to me which species combination and resource combination were chosen in the competition experiments. Fig S2 does not show the which are the novel conditions. Is is all. It will be a long list but could you provide a list of resourceXspecies combinations you tested?

Response: It is a fully crossed design so that providing the species combinations (Fig. S1) and resource combinations (Fig. S2) should be sufficient. We have clarified this now (L294-295). We have also made it clearer what are the novel conditions in Fig. 1b.

- line 274: Does it mean you randomly selected 8 out of 66?

Response: it is 8 out of the randomly selected 16 combinations. We have revised it (L291).

Fig1

- panel d: Again, explain the novel conditions.

Done

- panel e: Do the prediction accuracies of 3-sp and 4-sp communities differ from the 2-sp ones?

Response: No. We have clarified it (L111-112).

- What's the sample sizes of 2, 3, 4, 6-sp communities here for each density plot?

Response: We have added the sample sizes in the legend (now Fig. S6). There are 392, 196, 196, and 196 communities for 2, 3, 4, 6-sp combinations, respectively.

Fig2

- panel c-e. What does the "[frequency]" mean in the y axis? Does it mean the accuracy is computed based on BC similarity of species frequency? It's confusing.

Response: Yes, it is the BC similarity of species frequency. We have changed it to relative abundance and moved the figure to the supplement.

- panel d: Does a novel condition mean that the resource combination where the predicted community was assembled is not used in monoculture experiment at all? I would include these terms in figure 1 to improve clarity.

Done.

Supplements

- line 13: typo "affected". Also, what are the 4 combinations that are not significantly affected? Are they all pairs, 6-species community or what? Based on functional redundancy, I

would expect that those communities that are not affected by resource conditions are more likely to be initially diverse communities rather than pairs

Response: Three of the two-species and one of the three-species combinations were not significantly affected by resource environments. In three of these four cases, a 'superior' species was present – one that has lower requirement in each of the tested resources than its competitors. We speculated that diverse communities were unlikely to contain 'superior' species because there is a high chance that at least one competitor has a lower requirement for at least one resource. (L17-21 in the supplement).

- line 50: I would specify both c and s in this MM equation as these are two randomly drawn parameters later in the simulation

Done.

- line 186: typo. 0.8 for substitutable resources

Done.

- line 323: could you provide more details on how these cell features were measured? It'd be very helpful for people working on the the same system.

Response: We have added how mean and average autofluorescence intensity and cell length were calculated (L331-333 in the supplement).

- Fig S2: black dots instead of red dots?

Response: we have used black dots now.

- Fig S3. Not sure I understand the rationale behind PCA of beta-distance matrix. Why don't you used PCA directly on the relative abundance at 12day? Also, It'd be nice to see the final community richness/relative abundance for each culture in a bar chart. I wonder if the U-shape commonly seen in these PCA plots is the artifact of low diversity at the end of the experiments (at least it should be the case for combinations 1-16 where the initial communities are pairs)

Response: We have taken the suggestion, performing the PCA directly on the relative abundance. The conclusion holds. As we explained above, we did not show the final species richness because it did not differ with the initial richness in most communities. Because the competition experiments totalled 960 cultures, bar chart for each of them could be overwhelming. Instead, we have now plotted the final relative abundance for each species (Fig. S5).

Reviewer #1 (Remarks on code availability):

The scripts are not available for review.

Response: We have re-uploaded the scripts. Let us know if it does not work.

Reviewer #2 (Remarks to the Author):

Recently, many experiments have shown that resource competition is essential in the assembly of diverse microbial communities in complex environments by reproducing experimental observed patterns with consumer-resources models (Elife 11 (2022): e75168., Nature Microbiology 9.4 (2024): 1036-1048). Given the complexity of these interactions, coarse-graining approaches become necessary, as many models contain numerous underdetermined parameters.

In this study, the authors focus on a simpler, controlled system—monoculture experiments with green algae. They determine parameters for consumer-resource models (CRM) from experimental data and directly evaluate Tilman's rules of competition from the models, offering a fresh perspective on resource substitutability and its role in promoting diversity.

Here are my major comments:

1. The authors emphasize the well-known limitations of Lotka-Volterra (LV) models. However, they could strengthen their argument by focusing more on the experimental system used, especially how the plankton ecosystems compare with bacteria ones. Because of cross-feeding, microbial systems tend to have complex interactions, resulting in difficulties in precise modelling. Are cross-feeding effects important in green algae? If resource competition indeed dominates, this system could provide a more ideal framework for CRM theory validation. It would be nice to have some discussions on this point.

Response: we have introduced (L52-57) and discussed the difference between plankton and bacteria (L199-201).

2. Line 39, the authors used “phenomenological approach”. It seems they may have adopted this terminology from Tilman's original paper, but the term 'phenomenological' may be misleading. In this paper, the authors used a reasonable model (CRM instead of LV) but the approach is still “phenomenological” as it still uses models to explain experimental phenomenon. I recommend deleting the term “phenomenological” for clarity and precision.

Response: we have now deleted “phenomenological” throughout the MS.

3. The manuscript lacks direct comparisons between experimental data and model simulations, which weakens the overall argument.

3.1. In Fig. 1c, 2a and others, the absence of direct comparisons between experimental data and model outputs makes it difficult to assess the accuracy of the inferred parameters. Including the original data points in these figures would significantly improve their interpretability and allow for a more thorough assessment of model fit.

Response: We have now visualized the growth rates vs. initial resource concentrations or modelled resource concentration for one species under different concentrations of nitrate (Fig. 2a). We believe that these figures will provide a more direct assessment of model fit and explain the novelty of our approach to the experts. We include all the species per resource in the supplement (Fig. S3).

3.2. In Fig 2 c-e, the authors use Bray-Curtis similarity as the metric for accuracy. However, given the small number of species involved, this may not be the most appropriate choice. A direct comparison of community composition between simulations and experimental results could offer more insight into model performance.

Response: We have now directly compared the observed vs. predicted (simulated) relative abundance (Fig. 3). The results are qualitatively similar with those of using Bray-Curtis similarity. Because the observed-v.s.-predicted-abundance approach does not statistically tell whether the consumer-resource model is better than the null model, we kept the use of Bray-Curtis similarity to compare the predictive accuracies between models and between conditions.

3.3. In Fig 3, While I understand the intent behind Figure 3, the two vertical lines drawn on the probability density may confuse readers as they represent distinct aspects. I suggest moving some theoretical results from the SI to the main text and then using the probability density to reinforce those theoretical conclusions would be more helpful.

Response: We had added text on Fig. 3, pointing out that the vertical lines are calculated from the experiment and that the density plots are theoretical results (simulation).

3.4. In Fig. 4, the distinction is notable only at Richness=6. I suggest the authors further explain this inconsistency with theoretical predictions. One possible explanation is that the real feasible region for coexistence is not as large as the angle it spans due to undersampling issues. Alternatively, the authors may wish to discuss if it results from the limitations of the experiments, as mentioned in the Methods section. A deeper analysis and explanation of this inconsistency would enhance the manuscript.

Response: We discussed several points that may lead to the discrepancy between the experiment and the theory, including the complexity of nitrate usage, storage of phosphorus, and presence of neutral coexistence (L180-192).

While the conclusions and theoretical analysis presented in this paper are promising, the manuscript focuses primarily on predictions from consumer-resource models in simple ecosystems. As such, it is crucial to achieve stronger self-consistency and alignment between the models and experimental results, particularly in comparison to similar studies on more complex environments. Unfortunately, the current manuscript lacks sufficient direct comparisons between experiments and simulations to validate the robustness of the models. Furthermore, some of the conclusions drawn appear weakly supported by the available data. Therefore, I am unable to recommend this manuscript for publication in its present form.

Response: We have now directly compared the data and models for the monoculture and competition experiments and have discussed the discrepancy between the experiments and theory. We hope these have addressed the concern of the reviewer.

Minor comment:

Caption of Figure S2 in SI: BLACK dots indicate the competition experiments. Or you may redraw the figure with red dots to be consistent with Figure S18.

Done.

Reviewer #3 (Remarks to the Author):

MacArthur's consumer-resource model (CRM) has become a staple of modern ecological theory, serving as one of the primary vehicles for building intuition about the factors that affect species coexistence. It has the great merit of being the simplest model that respects the most basic physical constraint on population dynamics: a population can expand only by converting materials from its environment into biomass. The CRM achieves its simplicity by assuming that the availability of these resources is the dominant factor in determining population growth rates.

Despite the importance of the CRM for theoretical ecology, it has proven remarkably challenging to explicitly test experimentally. Ideally, one would begin by testing the model on microbial populations, where the experimental conditions can be precisely controlled, and the spatial and temporal scales allow for many replicates in a study of reasonable duration. In these systems, however, the resources are generally small molecules, whose abundances are barely detectable in the regime of resource-limitation where the CRM is most likely to be valid.

In this manuscript, the authors seek to begin addressing this lacuna, by testing a generic and ecologically meaningful prediction of the CRM on a large set of combinations of species of algae. Specifically, they focus on the stability conditions for the two-species fixed point of CRM dynamics: under a biologically plausible parameterization, these conditions are usually satisfied in competition for substitutable resources, while they are harder to guarantee in the case of essential resources. The authors quantify this theoretical claim using a specific implementation of the CRM with randomly sampled parameters, showing that the probability of stability is indeed much lower for essential resources.

To test this prediction, the authors estimate the CRM parameters from monoculture experiments with known initial resource concentrations. They face the same obstacle as prior efforts to test the CRM, and are unable to detect the resources when their concentrations approach the level where they begin to limit growth. But they circumvent this difficulty by performing a global fit of the model to all the monoculture experiments performed with the same species in various starting concentrations of the same resource. The variety of initial conditions compensates for the absence of full time-series data for the resources.

With such a well-posed and relevant question and a sound experimental method, this manuscript should eventually be published in a journal like Nature Communications. In my view, however, the results deserve further analysis before the article is released. The immediate message of Figures 3 and 4 is much more interesting than the authors' discussion would indicate, since the data clearly contradict the theoretical expectations. Whereas stable coexistence was supposed to be much more difficult for essential resources, the inferred parameters satisfy the stability conditions in about half the combinations for both kinds of resources, and the distribution of final diversity in the multi-species experiments in the two cases is almost indistinguishable. For me, this unexpected result is the principal message of the article, opening up promising possibilities for further investigation. To finish the project, the authors should at least propose one or two plausible hypotheses to explain the contradiction, possibly involving a few additional experiments to determine which is most viable.

Response: Thanks for the kind summary of our MS. We have provided several hypotheses on the discrepancy between the experiments and theory, including the complexity of nitrate usage, storage of phosphorus, and presence of neutral coexistence (L189-192).

My own hunch is that the problem lies in the assumption of independent random parameters in the probability calculations. At the most basic level, the function that maps resource concentration to population growth should include a scaling factor that depends on the species' cell size, as the nutrient uptake rate scales with cell surface area while the resource requirements scale with volume. This will introduce correlations among the parameters for a given species that will alter the probability calculus. If this is insufficient to explain the data, it may be worth investigating the biological factors that determine net uptake stoichiometry in more detail, to see whether physical constraints perhaps force the impact vectors to be sufficiently similar as to admit a metastable equilibrium.

Response: We have explored this question with the following model (linear version):

$$\frac{1}{N_i} \frac{dN_i}{dt} = \frac{c_{ij} w_{ij} d_i^2 R_j}{d_i^3} - m = \frac{c_{ij} w_{ij} R_j}{d_i} - m \quad (1)$$

$$\frac{1}{N_i} \frac{dR_j}{dt} = a_j(R_j) - c_{ij} d_i^2 R_j \quad (2)$$

where d_i is the cell size (e.g. diameter) of species i . c_{ij} and w_{ij} are constants that describe consumption rate and conversion rate of species i on resource j , respectively. We can see that nutrient uptake rate (i.e., consumption rate) scales with the area (d_i^2 in equation 2) and the requirement scales with the volume (d_i^3 in equation 3). $a_j(R_j)$ is the resource supply rate.

To assess the rule one, each species must be limited by different resources, we first calculate the R^* :

$$R_{ij}^* = \frac{d_i m}{c_{ij} w_{ij}}$$

Then, we assess whether

$$R_{11}^* < R_{21}^* \ \& \ R_{12}^* > R_{22}^* \ \text{or} \ R_{11}^* > R_{21}^* \ \& \ R_{12}^* < R_{22}^*$$

Because d_i is in the R^* , the result did differ with that of the main text, with 39.2% (was 50%) of the simulated species pairs meeting the rule one.

To assess the rule two, each species must consume more of the resource that more limits itself, we calculate the per capita consumption rate of each species on each resource. Species 1 consumes more of resource 1 than species 2, if:

$$\frac{c_{11} d_1^2 R_1}{c_{12} d_1^2 R_2} > \frac{c_{21} d_2^2 R_1}{c_{22} d_2^2 R_2}$$

We can see that the d_i cancels out, meaning that the scaling of size did not affect Rule Two.

In summary, while incorporating the scaling of cell size affects rule one, it affects competition for both essential and substitutable resources in the same way. Additionally, it does not affect rule two. Therefore, it does not explain why we observed a smaller effect of resource type in our experiment. While we agree that including scaling factors and/or parameter correlations could help explain the discrepancy between theory and experiment, we believe this would require a separate study. However, if the reviewer recommends including this analysis in the current study, we would be happy to do so.

Specific comments:

1. Figure 1a is confusing, especially the top panel where it seems like the “superior” species is not limited by any resource, but I don’t see any obvious way of improving it. Since the paper is about Tilman’s rules, one could simply begin immediately with ZNGI’s and impact vectors. This may be more difficult for some readers to follow, but would be useful for those most interested in the result.

Response: We considered beginning with the ZNGI figures in earlier version of the manuscript, but decided to use the current one for more general readership. We have now directed the specialist to the corresponding ZNGIs in Fig.1 and added the illustrations to the side of the ZNGIs in the supplement for comparison (Figs. S9 & S11).

2. In Figure 2, I would like to see at least one example of the measured per-capita growth rate vs. (inferred) resource concentration as compared with the Monod fit, and not be forced to re-plot the data myself from the repository in order to get a feel for what the actual measurements look like. Perhaps the authors could select two or three representative species and plot the measurements together with the fits, and save the rest of the curves for the supplement.

Response: We have taken the suggestions (Fig. 2a & S3).

3. The model without resource consumption (as in Figure 2c) seems hard to justify as a meaningful point of comparison, since in this model all the population densities are forever expanding exponentially, quickly reaching physically impossible levels. This strangeness is hidden by the use of relative species frequencies in place of absolute population sizes, but that does not change the unphysical character of the model. The authors should eliminate this comparison, or devise another way of testing the relevance of the impact vector.

Response: We have deleted the model without resource consumption.

4. On page 8, line 159, the authors relate their results to the paradox of the plankton. Most discussions of the paradox of the plankton focus on Tilman’s first rule, noting that the intersection of all the ZNGI’s demands fine-tuning of parameters when the number of species exceeds the largest plausible resource dimension. Since this article concerns the second rule, the reference requires more explanation, or should be eliminated.

Response: We have deleted this part.

REVIEWER COMMENTS

Reviewer #1 (Remarks to the Author):

The authors have addressed all the comments I raised and greatly improved the clarity of the manuscript. I am generally satisfied with the revisions made. Here are some minor comments for further consideration:

- Methods and Supplements. Please use mL instead of ml.

Done.

- Lines 232 and 234. Please remove -N after nitrate and ammonium.

Done.

- Line 238. "All species and resource concentrations were randomized." I thought it's a fully crossed design? Or are they randomized on the 96-well plate?

Response: We meant that they were randomized on the plates, which we have clarified now (L241).

- Line 253. Is mortality rate really dependent on the resource? If so I think it should mean mortality in the presence of resource. If not, it should be just m_i .

Response: We have changed it to 'in the presence of resource' (L256).

- Fig2 and S2. I recommend moving the point legend outside of the panel. It's somewhat confusing to have open legends within the panel.

Done.

- FigS3 caption. Typo modedlled

Done.

- Fig S6. Regarding the sample sizes for 2, 3, 4, 6-species communities, I think it's 384, 192, 192, 192 which adds up to a total of 960.

Response: Yes, we have changed it.

Reviewer #1 (Remarks on code availability):

Yes, a README file and main data table are included. The Rmd and python scripts are self-describing. I would appreciate including the scripts for the image processing pipeline as it will hugely benefit researchers working on similar platforms.

Response: We have now uploaded all the data and scripts in the Figshare.

Reviewer #2 (Remarks to the Author):

My concerns have been well addressed.

Response: We thank the reviewer for the valuable suggestions.

Reviewer #3 (Remarks to the Author):

The authors have taken significant steps to address all my concerns from the first report. I do recommend including the cell-size scaling factor analysis in the current study, at least in the supplementary material, with a mention in the main text. The fact that the rule 1 probability drops to 40% -- very close to the substitutable-resource experimental value of 38% -- removes the open question about the source of the hidden correlations among parameters in the experiment, and helps strengthen the main result.

Response: We have now discussed it (L184-186) and presented the details in the supplement S7.

It would also be useful to compute how the probabilities change for rule 2. The authors claim that these probabilities are unaffected by the scaling factor, but they neglect the fact that these ratios still depend on the equilibrium resource concentrations, which are affected.

Response: Simulation also confirmed that the probabilities for rule 2 was unaffected by the scaling factor despite the change of equilibrium resource concentrations. However, the probability of coexistence did change because it is a product of rules 1 and 2. We have now discussed this (supplement S7).